# Localization and Mapping for Robots in Agriculture and Forestry: A Survey

**André Silva Aguiar** [1,2,*]**, Filipe Neves dos Santos** [1]**, José Boaventura Cunha** [1,2]**, Héber Sobreira** [1] **and Armando Jorge Sousa** [1,3]

1    INESC TEC—INESC Technology and Science, 4200-465 Porto, Portugal; filipe.n.santos@inesctec.pt (F.N.d.S.); jboavent@utad.pt (J.B.C.); heber.m.sobreira@inesctec.pt (H.S.); asousa@fe.up.pt (A.J.S.)
2    School of Science and Technology, University of Trás-os-Montes e Alto Douro, 5000-801 Vila Real, Portugal
3    Faculty of Engineering, University of Porto, 4200-465 Porto, Portugal
*    Correspondence: andre.s.aguiar@inesctec.pt

**Abstract:** Research and development of autonomous mobile robotic solutions that can perform several active agricultural tasks (pruning, harvesting, mowing) have been growing. Robots are now used for a variety of tasks such as planting, harvesting, environmental monitoring, supply of water and nutrients, and others. To do so, robots need to be able to perform online localization and, if desired, mapping. The most used approach for localization in agricultural applications is based in standalone Global Navigation Satellite System-based systems. However, in many agricultural and forest environments, satellite signals are unavailable or inaccurate, which leads to the need of advanced solutions independent from these signals. Approaches like simultaneous localization and mapping and visual odometry are the most promising solutions to increase localization reliability and availability. This work leads to the main conclusion that, few methods can achieve simultaneously the desired goals of scalability, availability, and accuracy, due to the challenges imposed by these harsh environments. In the near future, novel contributions to this field are expected that will help one to achieve the desired goals, with the development of more advanced techniques, based on 3D localization, and semantic and topological mapping. In this context, this work proposes an analysis of the current state-of-the-art of localization and mapping approaches in agriculture and forest environments. Additionally, an overview about the available datasets to develop and test these approaches is performed. Finally, a critical analysis of this research field is done, with the characterization of the literature using a variety of metrics.

**Keywords:** localization and mapping; autonomous navigation; agriculture; forestry

## 1. Introduction

There have been several developments in the research and applications of robotic solutions for the agriculture sector and novel contributions in the near future are expected [1,2]. The need for automatic machines in this area is increasing since farmers increasingly recognize its impact in agriculture [3]. Robots are now used for a variety of tasks such as planting, harvesting, environmental monitoring, supply of water and nutrients, and others [4]. In this context, developing solutions that allow robots to navigate safely in these environments is essential. These advances impose a main requirement: localizing the robots in these agriculture and forestry environments. The most common solution is to use Global Navigation Satellite System (GNSS) standalone-based solutions [5,6]. However, in many agricultural and forestry places, satellite signals suffer from signal blockage and multi-reflection [7,8], making the use of GNSS unreliable. In this context, it is extremely important to research and develop intelligent solutions that use different modalities of sensors, as well as different sources of input

data, to compute the robot localization. Simultaneous localization and mapping (SLAM) [9,10] is the state-of-the art approach to do so. This technique consists of estimating the state of a robot using input sensor data, while simultaneously building a map of the surrounding environment [11]. The robot model usually comprises its pose, and, in some cases, its velocity, calibration parameters, sensor offsets, among others. The map is a multi-dimensional representation of the agents observed by the on-board robot sensors, that are used as references in the localization procedure. The map creation is usually important to provide information about the environment. In agriculture and forestry, they can be used by human operators to report information about the cultures. Furthermore, maps can be saved and loaded, being reused and updated by the robotic platforms each time they operate in the terrain. When mapping the environment is not desired, alternatives to SLAM are also approached. One of the most common is visual odometry (VO) [12]. As stated by Scaramuzza et al. [13], VO is the process of estimating the motion of the on-board camera (s) using only image data as input.

Given all of the above, one can conclude that there is a huge dependency on GNSS-free localization systems from autonomous mobile robots working in agriculture and forestry. So, this leads to the main question: is robotics localization a solved topic? The answer to this question depends on many aspects: the context where the robot operates, the quality of the on-board sensors, and, the desire performance of the localization system. For example, 2D LiDAR-based SLAM in indoor environments is a mature research field, with many high-quality state-of-art methods [14,15] reporting high-performance results. On the other hand, SLAM in harsh outdoor environments (as agriculture and forestry) is still a growing research topic. These are highly dynamic environments that change drastically over the year, which makes long term mapping a difficult task. To overcome this, the concept of 4D mapping, i.e., spatio-temporal reconstruction of the environment [16], is becoming popular. Additionally, the characteristics of illumination and terrain irregularities lead to a more unstable motion and to a more difficult perception of the environment. Furthermore, both SLAM and VO suffer from the well known drift problem, and usually in these environments, it is intended that robots perform long term operations. So, as these localization algorithms tend to accumulate error over time, for long term operations the drift can be quite significant. To overcome this, many SLAM algorithms are endowed with the capability of recognizing previously visited places. With this, they can detect loop closures [17,18] and correct the drift issue.

Even though so many solutions have been proposed to solve localization and mapping main issues in agriculture and forestry, it is clear that there are still several working lines to be improved. The difficulty of this problem leads to the creation of new solutions and the development of new concepts to localize outdoor robots and make them autonomous. This work explores the research field of localization and mapping in agriculture and forestry, highlighting the solutions that are present in the literature to solve this problem. In a first stage, in Section 2, the general approaches to solve SLAM and VO are detailed, and the main issues inherent to them are described. In this, the solutions to solve the theoretical SLAM problem are detailed, the mapping approaches and data association problems are discriminated, and VO is also presented. Then, Section 3 presents the methodology used in this work to collect the set of works presented in the article, related to localization and mapping in agriculture and forestry. Following this description, Sections 4 and 5 present the respective works. Next are described the existing datasets that contain sensor data acquired in agriculture or forestry that can be used in localization and mapping. Finally, the main conclusions of this work are outlined in Section 7.

## 2. The Localization and Mapping Problem

Localization and mapping can be approached in several different ways. The most common technique is to perform both tasks simultaneously, as performed in SLAM. Additionally, localization can be performed singly. On one hand, maps previously built can be loaded and used to localize the robot inside it. On the other, localization can be performed without registering the observed keypoints, as in VO. This section presents a description of all these approaches.

*2.1. The SLAM Method*

SLAM was originally proposed by Smith and Cheeseman [19] in 1986. By this time, robotics started to be addressed in innovative ways, considering probabilistic viewpoints. This new way of thinking robotics lead to the consideration on how to manipulate geometric uncertainty [20], which was crucial to the development of mapping techniques. An example is [21] that came up with the concept of a stochastic map, that is a map as we know it by now, considering relationships between objects and their uncertainty, given the available sensor data information. All these concepts provided the sufficient knowledge to the creation of the well known SLAM problem: estimating the map objects location along with the robot pose in a single joint state [22]. So, from the very beginning, SLAM was formulated as a probabilistic problem, as will be described further.

2.1.1. Solving the SLAM Problem

The SLAM problem, as referenced in [23], can be defined as: an autonomous mobile robot starts its travel in a known starting point, and explores an unknown environment. The SLAM offers a solution to estimate the unknown robot motion while building a map of the unknown environment. This solution had been formulated statistically, in many different ways. Let us denote each time instant as $k$, the robot pose at each instant as $x_k$, and the full robot trajectory until the time instant $k$ as

$$X_k = \left\{ x_0, x_1, ..., x_k \right\}. \tag{1}$$

The robot pose encodes the robot location and orientation, and can describe both 2D and 3D spaces. Then, SLAM uses two sources of input information. The first are the controls $u_k$ that usually represent odometry, i.e., relative information from wheel encoders between two consecutive robot poses. Additionally, other sources of controls can be used such as inertial sensors. The historic of control inputs is here represented as

$$U_k = \left\{ u_0, u_1, ..., u_k \right\}. \tag{2}$$

The other source of input information are observations taken from on-board sensors $z_k$. With these measurements, a map $m$ is built using registration methods. Similar to robot states and controls, observations can be stored and their historic can be defined as

$$Z_k = \left\{ z_0, z_1, ..., z_k \right\}. \tag{3}$$

Figure 1 shows a graphical representation of SLAM, as well as all the previously mentioned variables involved in it.

In this, the relationships between the entities are well defined. It is possible to observe the sequence of robot poses, as well as the interconnection of the localization and mapping procedures.

Given all of the above, probabilistic SLAM can be addressed in two different ways. The first estimates the posterior distribution

$$p(X_k, m | Z_k, U_k, x_0), \tag{4}$$

i.e., the full robot trajectory and the environment map, given the entire set of observations, all the control inputs, and the first robot pose. The second approach does not consider the entire robot path, but only its current pose, as follows

$$p(x_k, m | Z_k, U_k, x_0). \tag{5}$$

This approach performs an incremental estimation and discards the historic of poses. To solve Equations (4) and (5), two main groups of SLAM approaches have been created and are still under development: filter-based SLAM and optimization-based SLAM.

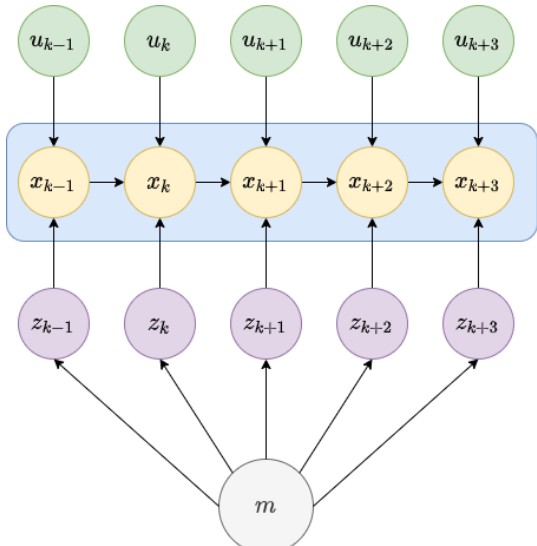

**Figure 1.** Graphical representation of simultaneous localization and mapping (SLAM). The robot moves from $x_{k-1}$ to $x_{k+3}$ through a sequence of controls. At each time instant, features are observed by the on-board robot sensors, and are registered on the global map $m$.

Filter-based SLAM is derived from the Bayesian filtering and is composed of two main steps: prediction step and update step. In the first, the robot pose is updated through a motion model that considers the control inputs. This is defined as

$$p(x_k|x_{k-1}, u_k). \tag{6}$$

With this information, the joint posterior can be updated integrating the motion model over the previously joint state as follows

$$p(x_k, m|Z_{k-1}, U_k, x_0) = \int p(x_k|x_{k-1}, u_k) \cdot p(x_{k-1}, m|Z_{k-1}, U_k, x_0) dx_{k-1}. \tag{7}$$

Similarly, the update step uses an observation model defined as the probability of making an observation $z_k$ given the robot pose and map configuration, as follows

$$p(z_k|x_k, m). \tag{8}$$

Usually, SLAM assumes that once the robot pose and map configuration are known, there is a conditional independence between observations. With this information, the update step is formulated as the following conditional probability

$$p(x_k, m|Z_k, U_k, x_0) = \frac{p(z_k|x_k, m) \cdot p(x_k, m|Z_{k-1}, U_k, x_0)}{p(z_k|Z_k, U_k)}. \tag{9}$$

With all these fundamentals, several filter-based approaches are used to solve the SLAM problem.

1. The extended Kalman filter (EKF): EKF-SLAM [24,25] is based on the well known group of Kalman filters (KFs) [26]. Since both SLAM-based motion and observation models are, in most cases non-linear, the original KF is not suitable for such systems. In this context, EKF is usually used, since it linearizes the models. Additionally, this approach considers that the state uncertainty is approximated by a mean and a covariance, i.e., a Gaussian distribution. EKF-SLAM state is formulated as $f(x_{k-1}, u_k) = [x_k, m]^T$, i.e., the robot pose and the map built so far. This means that the motion model is described as

$$p(x_k|x_{k-1}, u_k) = f(x_{k-1}, u_k) + w_k, \tag{10}$$

where $f(.)$ is a non linear representation of the robot motions, and $w_k$ represents Gaussian motion noise with covariance $\mathbf{Q}_k$. This formulation can lead to a high-dimensional filter, as long as the map grows, that can have serious impact on the SLAM algorithm runtime performance. Since the SLAM update time depends quadratically on the size of the state vector, EKF-SLAM is usually not suitable for large scale environments. To solve this issue, some approaches developed the sub-map concept [27,28]. In these, the global map is partitioned in local sub-maps, that can share information, still becoming conditionally independent. In this way, the state vector dimension is reduced, and the global map can still be recovered by the composition of the local sub-maps. To introduce the observations, EKF-SLAM formulates the observations model in a similar way of what is done in the motion model, by linearizing a non linear function $h(x_k, m)$ that is introduced in the model as follows

$$p(z_k|x_k, m) = h(x_k, m) + v_k, \tag{11}$$

where $h(.)$ describes the observations and $v_k$ is zero mean Gaussian noise with covariance $\mathbf{R}_k$. Both $f(x_{k-1}, u_k)$ and $h(x_k, m)$ are formulated accordingly with the input data and the type of mapping approach used.

2. The information filter (IF) in the IF [29], also known as inverse covariance filter, the covariance matrix is replaced by the information matrix, i.e., its inverse. In comparison with the KF and EKF, in case of complete uncertainty, i.e., when the covariance is infinity, the information matrix is zero, which is easier to work with. Additionally, the information matrix is usually sparser than the covariance matrix. On the other hand, if the system presents non-linearities, to recover the mean is required to solve a linear system of equations. In the literature, IF is not as popular as the EKF to solve the SLAM problem. Even so, some works adopt this approach [30,31], and it is especially popular in multi-robot SLAM systems [32].

3. The particle filter (PF): PFs are based on Monte Carlo sampling, and are widely used in the SLAM context. In these, the system state is sampled by a well-defined number of particles with a given likelihood. each particle encodes information about the robot pose, and can also contain the map data. The PF overcomes the limitation of KF-based approaches, by not restricting the motion and observation models noise to zero mean Gaussians. This formulation can approximate the on-board sensors characteristics in a more realistic way. In this context, FastSLAM [33] was a huge mark in probabilistic SLAM research. This algorithm considers $N$ particles where each one contains the robot trajectory $X_k$ and a set of 2-dimensional Gaussians representing each landmark on the map, solving the full-SLAM problem represented in Equation (4). However, the high dimension that characterizes SLAM systems can lead PFs to become computationally unfeasible, as the number of required particles increases to perform a good approximation of the state-space. To overcome this problem, the Rao–Blackwellization became popular in the SLAM context, where the Rao–Blackwellized PFs gained impact [15,34,35]. In these, the joint state is represented as $\{w_{k-1}^{(i)}, X_{k-1}^{(i)}, p(m|X_{k-1}^{(i)}, Z_{k-1}^{(i)})\}_i^N$, where $w_{k-1}^{(i)}$ is the weight of the *ith* particle. To solve the probabilistic SLAM problem, this approach factorizes the posterior distribution in the following way

$$p(X_k, m|Z_k, U_k, x_0) = p(m|X_k, Z_k) \cdot p(X_k|Z_k, U_k, x_0). \tag{12}$$

This formulation is the key factor that speeds up the filter, since the posterior over maps $p(m|X_k, Z_k)$ can be solved analytically given the robot path and the observations until the current instant.

4. Graph-based optimization SLAM: Graph-SLAM [36] is a whole different way of looking and approaching the SLAM problem, comparing to filter-based techniques. The basic principle of this approach is as follows: the robot and map features can be represented as nodes in a graph-way procedure. Then, arcs exist between consecutive robots poses $x_{k-1}$, $x_k$ representing the information given by the input controls $u_k$, and between robot poses and map features. Figure 2 represents this configuration.

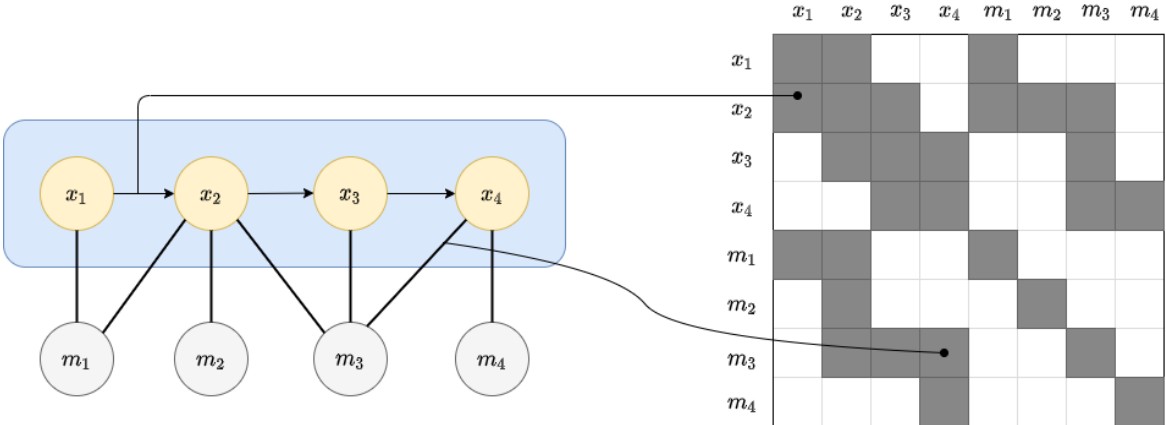

**Figure 2.** Illustration of graph construction present in Graph-SLAM. Adapted from [23].

In this figure, it is possible to observe that, for example, when the robot in the pose $x_1$ observed a map feature $m_1$, an arc is added between these two agents. This relation is also encoded in a matrix-like representation, where a value is added between the $x_1$ and $m_1$ components. When the robot moves (Figure 2b), the control input $u_2$ leads to an arc between $x_1$ and $x_2$. As the input data increase, the graph representation also increases. Even so, this representation is considered to be sparse, in that each robot pose node is connected to few other nodes. This leads to a great advantage in comparison with EKF-SLAM, since the update time of the graph is constant, and does not grow quadratically as in EKF-SLAM. As stated in [37], this graph-based approach can solve for the robot pose and map structure, by finding a minimum of a cost function of the following form

$$F(X_k, m) = \sum_{ij} e_{ij}(X_k, m)^T \, \Omega_{ij} \, e_{ij}(x_k, m), \tag{13}$$

where $X_k$ is the vector containing the robot trajectory until the time instant $k$, $m$ is the map, $e_{ij}$ is an error function that relates the prediction and the observations, and $\Omega_{ij}$ is graph-based matrix representation referenced before. Thus, the optimal values $(X_k^*, m^*)$ are obtained minimizing as

$$(X_k^*, m^*) = \underset{X_k, m}{argmin} \, F(X_k, m). \tag{14}$$

To solve (14), the state-of-the-art methods can be used, such as Gauss–Newton, Levenberg–Marquardt, and others.

### 2.1.2. Mapping the Environment

The mapping procedure is crucial in the localization performance. The accuracy of the on-board sensors, and the quality of the data post-processing algorithms dictate the quality of the perception of the surrounding environment by the robotic platform. In agriculture, this is specially true, since in most cases, the environment present harsh conditions for robotics localization and mapping. Figure 3 shows an example of a working place of an agricultural robot.

In this are visible the long corridors that constitute the vineyard, as well as the absence of structured objects, such as walls. All these characteristics complicate the mapping procedure. To build precise maps of these environments, the robotic platform present in Figure 3 uses high-range 3D LiDARs and stereo cameras. In this way, 3D dense reconstruction with color information is achieved using the vision systems, and high range omnidirectional maps are constructed using the 3D LiDAR sensors. To address the mapping procedures present in the SLAM context, in this article we start by describing the well known problem of data association. Then, the mapping approaches are divided in two main sets: metric maps, topological maps, and semantic maps.

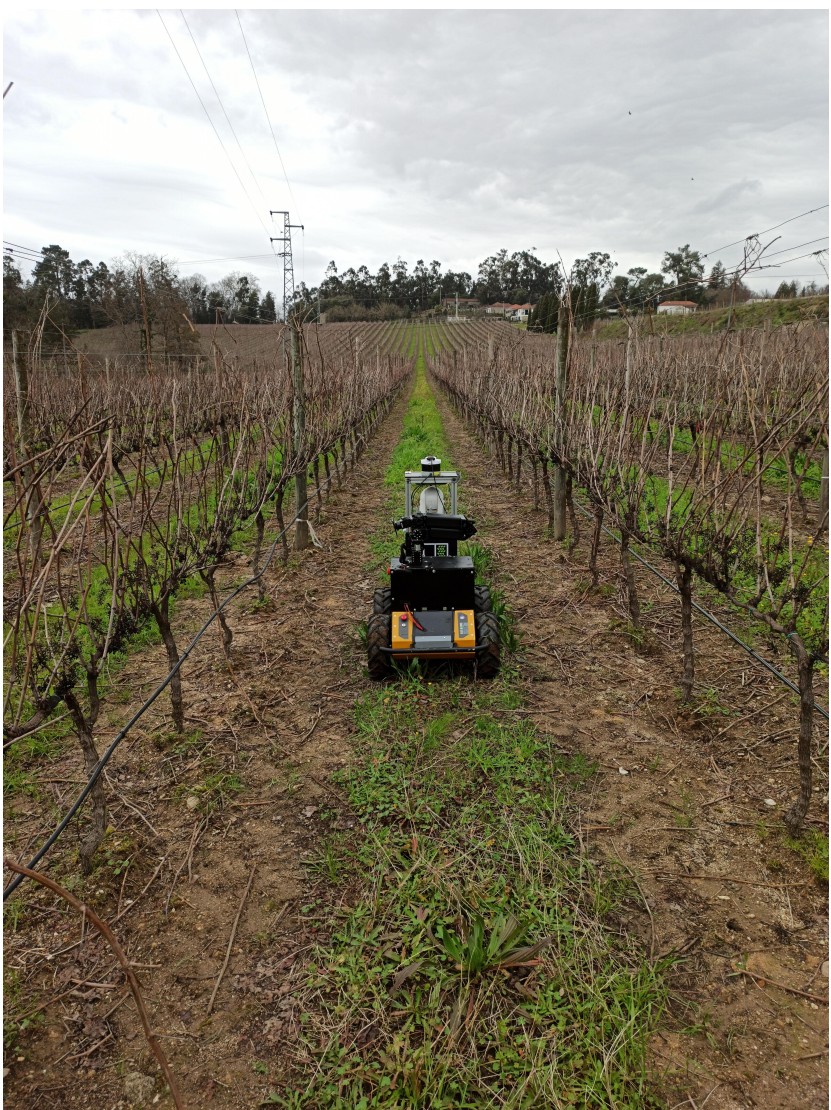

**Figure 3.** Agricultural robot working place on an agricultural environment [38]. The image shows the complexity of the mapping procedure, and the importance of the projection of an adequate system of sensors.

1. The data association problem: The data association in SLAM [39–41] is the process of associating a sensed feature in a given time instant, with another one observed in a different instant, ensuring that both correspond to the same physical point in the world scene. This procedure can constitute a problem in SLAM, since the number of possible associations between features grow exponentially over time, since the map becomes larger at each observation [42]. If data association is unknown, the estimation of the posterior distribution has to consider a high density of modes, which can lead to an infeasible computational cost. Moreover, the detection of loop closures is essentially a data association problem, in that associations between the global map and a local map are searched to find previously visited places by the robot.

The data association problem has been addressed in several works present in the literature [43,44]. The most common procedure is to perform incremental maximum likelihood association, i.e., choose the most likely association hypothesis. Other algorithms support the multi-association hypothesis, where in cases of high association uncertainty, a feature can have multiple associations until one of them has a higher likelihood. This, besides improving the success rate of data association, can lead to the growth of the number hypothesis exponentially [45]. In more advanced approaches, machine learning is used to compute data association algorithms, performing decisions for groups of features at once [46]. In general, considering its importance, data association is a topic which is always under research and improvement, considering new types of sensors, data, and features.

2. Metric maps: The metric representation is a structure that encodes the geometry of the environment [11]. Metric maps can be represented in several ways, depending on the input sensor data, and the dimension of the mapping procedure. One of the most common approaches is landmark-based mapping procedures. In these, the environment is represented as a set of 3D landmarks, in a sparse way. The landmarks encode relevant features of the scene, such as planes, lines or corners [47], and are represented as unique signatures (also called descriptors). A good descriptor is crucial for the proper performance of the mapping procedure. The more unique is the descriptor, the easier is the data association procedure. Additionally, metric maps can represent the scene in a more structured way, using occupancy grid maps. Typically, these maps represent the environment in 2D, and sample the geometric space into cells with an associated probability of occupation [48].

3. Topological maps: The concept of partitioning the geometric space into local maps gained strength in the SLAM mapping procedure. In this context, topological maps come up as a logical solution. These algorithms represent the global map as a set of connected local maps stored in nodes [49]. As stated by Lowry et al. [50], a topological map is conceptually a biological cognitive map, where nodes represent possible places in the world and edges the possible paths that connect these nodes. In the SLAM context, the routes between nodes represent the robot navigation path. This context has the advantage of storing local maps in nodes, allowing the robot to load only the local maps of interest at each point in time. In agriculture, this is specially interesting, since usually maps are dense and long-term, leading to high memory consumption. The partitioning of the map allows greater memory efficiency, in that only portions of the map memory are loaded at each instant. On the other side, this technique can have implementation issues. For example, the robot has to be able to associate physical places and routes to nodes and paths. Moreover, the extraction of the topological map can also be challenging, and usually needs the definition of a metric occupancy grid map of the environment.

4. Semantic maps: In many cases, adding semantic information to maps can enhance the quality of the localization systems [51]. This is a whole new challenge for perception systems, that have to be able to recognize and classify objects and places. To build a semantic map, several aspects have to be taken into account. For example, the detail of semantic concepts is important, and depends mainly on the robot task. If the robot wants to move from some corridor into another, the semantic classification can be as high-level as recognizing corridors. On the other hand, if the robot is intended to harvest, the semantic classification should be able to recognize trunks, leaves, etc. Another important concept is the definition

of the semantic properties, since a single object can be characterized by an unlimited number of concepts. This process is crucial for the mapping procedure, and can be viewed as a dictionary of the environment.

5. Hybrid maps: All the previously described maps can be merged and fused, creating the so-called hybrid map architectures. For example, objects or places can be semantically characterized and represent nodes of a topological map that holds local metric maps. For example, in [52] is proposed, a probabilistic map representation that considers objects and doors, and classifies places using prior information about the objects. Figure 4 shows another representation of an hybrid map, applied to an agricultural context.

In this, the vineyard is semantically characterized as rows and corridors, in a topological map that has paths interconnecting each node.

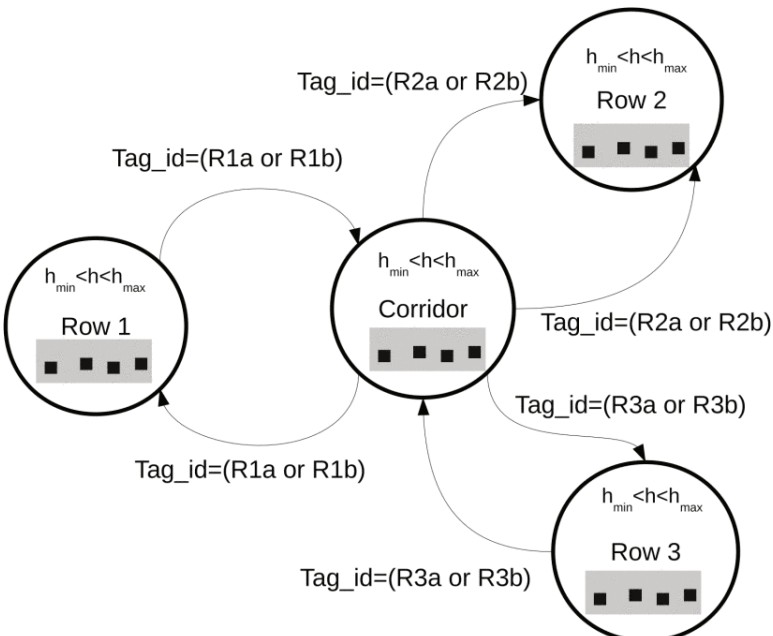

**Figure 4.** Semantic and topological map of a vineyard [53]. Nodes represent well defined places and contain semantic information: they either represent a vineyard row or corridor. Local maps of artificial landmarks are stored in each node. Edges encode the geometric path between nodes.

*2.2. The Visual Odometry Method*

VO's ultimate goal is to localize the absolute camera pose in relation with a given initial reference pose, as represented in Figure 5.

Contrary to the SLAM problem, VO does not maintain a map of the environment. This approach receives a set of consecutive images, computes the relative transformation between them, and integrates each transformation to recover the absolute camera pose. This algorithm is a particular case of structure for motion, that solves the problem of 3D reconstruction of the environment and the camera poses using a sequence of unordered image frames [54]. To compute the relative transformation between images, VO is composed of a sequence of steps. Firstly, the input image is processed so that features can be extracted from it [55]. This is done so that robust feature matching algorithms can be executed between images. Then, matches between features are used to estimate the motion between images. This motion estimation can be computed in several ways, such as 2D to 2D, where the features are specified in 2D coordinates for both images. Moreover, 3D to 3D motion estimation can be calculated, using depth information of stereo cameras, specifying the images features in the 3D space. In some cases, bundle adjustment [56] techniques are applied to refine the local estimate of the trajectory. This method is implemented aiming to optimize the camera parameters and the 3D landmark parameters simultaneously.

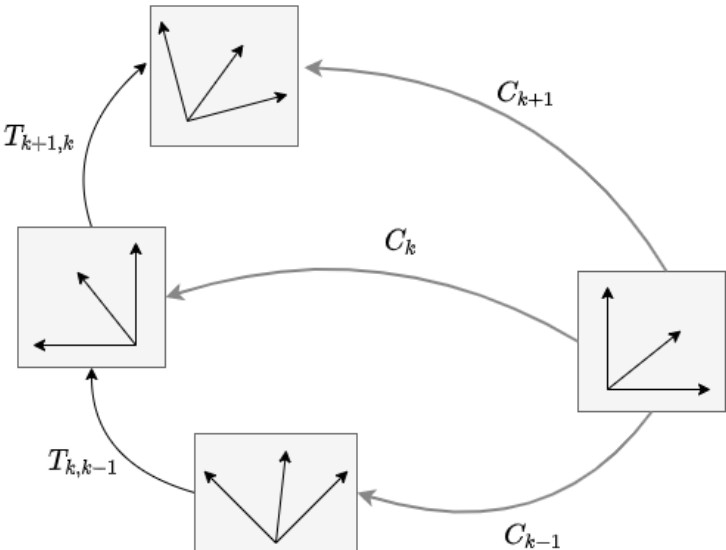

**Figure 5.** The Visual Odometry problem. Just like in wheel odometry, each relative transformation between camera poses $T_{k,k-1}$ is integrated in order to recover the absolute camera pose $C_k$, regarding an initial coordinate frame. Adapted from [13].

## 3. Methodology

The previous sections provided an overview of the variety of methods and solutions for robotics localization and mapping. This work pretends to evaluate the performance of these approaches in agricultural and forest environments. The main goal is to understand if this is still an open issue in the scientific community, and, if so, what are the target research areas to overcome the limitations of the science in this field. To do so, a deep literature analysis was performed. The result was the collection of 15 works on agricultural fields, and 9 on forest environments. To evaluate them, the following criteria were considered:

- *Application*: Agricultural or forest application of the desired autonomous system.
- *Localization approach:* The methods and sensors used to localize the robot.
- *Mapping approach:* The methods and sensors used to map the environment.
- *Accuracy:* Evaluation of how accurate the robot localization is. In the ideal case the accuracy show be less that some value (usually 20 cm [37,57]).
- *Scalability:* Evaluation of the capacity of the algorithm to handle large-scale paths.
- *Availability:* Evaluation of the possibility of the algorithm to present reliable localization right away, without need for building prior maps of the environment.

Given the immature status of this research field, let us not consider more advanced metrics such as recovery, updatability, or dynamicity [37]. Figure 6 shows the distribution of the years of creation of the works collected.

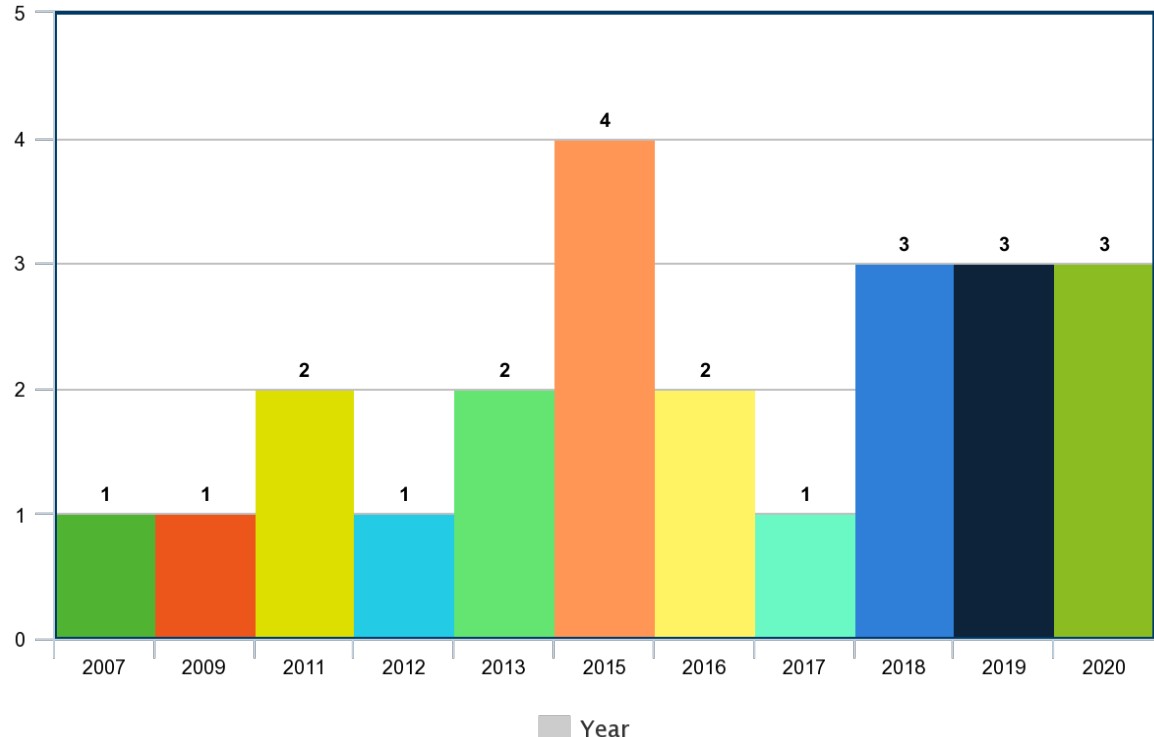

**Figure 6.** Histogram representing the years of creation of the works collected of localization and mapping in agriculture and forestry. The great majority of them were developed since 2011.

From this, it is possible to observe that this research field is still recent, since the majority of works were developed since 2011. Finally, dataset collections in agriculture and forest for autonomous driving were also searched in the literature. This resulted in the collection of 5 works. To evaluate them, the following topics were considered:

- *Description:* Agricultural or forestry area where the data was collected, as well as sensor information.
- *Large-scale:* Whether or not the data were collected in large-scale environments, and large-scale paths.
- *Long-term:* Whether or not the data were collected in different seasons of the year, and different times of the day.

It is worth noting that all these works were developed after 2016.

## 4. Localization and Mapping in Agriculture

Recent works approach the problem of localization and mapping in agriculture. This work performed an intense literature review in this area. Table 1 presents a summary of the works collected and their main characteristics.

**Table 1.** Summary of the works collected on localization and mapping related to the agricultural field.

| Ref. | Agricultural Application | Localization Approach | Mapping Application | Tested in Real Scenario | Accuracy | Scalability | Availability |
|---|---|---|---|---|---|---|---|
| Freitas et al. [58,59] (2012) | Precision agriculture in tree fruit production | (2D) EKF-based. Wheel odometry and laser data. Uses points and lines features to match witha previously built map. | Offline orchard metric mapping for artificial landmark detection. | **Yes**. Three experiments with more than 1.8 km each. | Low error for flat and dry terrains ( 0.2 m). High errors in steep terrains (up to 6 m). | **Yes**. Long-term experiments performed. | **No**. The method requires a previously build map. |
| Duarte et al. [60] (2015) | Autonomous navigation on steep slope vineyards. | (2D) PF-based. Fusion of GPS, wheel odometry, and previously mapped landmarks. | Offline metric mapping. Use of wireless sensors to compute landmarks location. | **Yes**. However, tests were done in an urban environment. | Beacons mapped with 1.5m of average error. Robot pose estimation not evaluated quantitatively. | **Not tested.** | **No**. No proper performance without a previously built metric map. |
| Zaman et al. [61] (2019) | Precision agriculture. | (2D) VO algorithm based on a cross correlation approach. | - | **Yes**. Tested in soil, grass, concrete, asphalt, and gravel terrains. | Normalized cumulative error of 0.08 mm for short paths. | **No**. System performance degrades when incrementing path lenght. | **Yes**. No need for first passage of thealgorithm in the agricultural field. |
| Habibie et al. [62] (2017) | Monitoring of ripe fruit. | (2D) Use of the state-of-the-art Gmapping [15] and Hector SLAM [14] approaches. | Combination of metric maps. Occupancy grid map generated by the SLAM approach, and fusion with tree/fruit detection. | **No**. Experiments only performed in simulation. | Localization not quantitatively evaluated. Accurate detection of simulated fruits, and trees. | **Not tested.** | **Not tested.** |
| Younse et al. [63] (2007) | Greenhouse spraying. | (2D) VO algorithm. Use of Kanade-Lucas-Tomasi (KLT) Feature Tracker. | - | **Yes**. Short-term tests in indoor environments, and outdoor with different ground surfaces. | 12.4 cm translation error for a short path of 305 cm, and 8° orientation error for a 180° rotation. | **Not tested.** | In theory. |
| Bayar et al. [64] (2015) | Autonomous navigation in orchards. | (2D) Fuses wheel odometry and laser range data. Assumes that orchards rows length is known. Localization only relative to the rows' lines. | - | **Yes**. Experiments performed in several orchards rows. | Low errors relative to the rows' lines. No quantitatively average values provided. | **Partially**. Only under the assumption that the row length is known, and the localization is relative to the rows' lines. | **No**. Requires the previously mentioned assumptions. |
| Le et al. [65] (2018) | General agricultural tasks. | (3D) Localization based on non-linear optimization techniques. Uses the Lavenberg-Marquardt algorithm. | 3D LiDAR mapping. Uses edge and planar features extracted directly from the point cloud. | **Yes**. Tested both on real and simulated scenarios. | Less than 2% translation error for a 500m trajectory. | **Yes**. A successful long-term experiment was performed. Loop closure supported. | **Yes**. The system is able to perform online SLAM without any priormap. |
| Cheein et al. [66] (2011) | Autonomous navigation on olive groves. | (2D) Extended IF-based SLAM. Uses a laser sensor and a monocular vision system. | Metric map composed of olive stems. Support Vector Machine used to detect stems, and laser data to map them. | **Yes**. Tests performed in a real olive scenario. | Successful reconstruction of the entire olive. Consistent SLAM approach. Error does not exceed 0.5m. | **Yes**. The method is able to be long-term consistent operating in the entire olive. | **Yes**. The system is able to perform online SLAM without any prior map. |
| Chebrolu et al. [67] (2019) | Precision agriculture in crop fields. | (2D) PF-based. Fuses wheel odometry with camera visual data. | Offline mapping. Metric-semantic map of landmarks build from aerial images. | **Yes**. Real experiments performed on sugarbeet field. | Maximum error of 17 cm on a >200 m path. | **Yes**. Experiments show good performance on long-term paths. | **Partially**. Only if a previously extracted aerial map is available. |
| Blok et al. [68] (2019) | Autonomous navigation in orchards. | (2D) PF-based: uses a laser beam model. KF-based: uses a line-detection algorithm. | - | **Yes**. Tests on two real orchard paths. | Lateral deviation errors <10 cm and angular deviation <4°. | **Yes**. Successful tests in orchards paths in 100 m. | In theory. Does not need any prior mapping information. |
| Piyathilaka et al. [69] (2011) | General agricultural tasks. | (2D) EKF-based. Fusion of a VO approach with a stereo vision range measurement system. | - | **Yes**. Experiments performed in real outdoor environment. | Average error of 53.84 cm on the tested sequence. | **No**. High cumulative error for long paths. | In theory. Does not need any prior mapping information. |
| Iqbal et al. [70] | Phenotyping, plant volume and canopy height measurement. | (2D) EKF-based. Fusion 2D laser, an IMU, and GPS data. | 2D point cloud map built by successive registration. | **No**. Experiments performed in the Gazebo simulator. | 2.25 cm error on 32.5 m path, and 7.78 cm on 38.8 m path. | **Not tested.** | **Yes**. Does not require prior mapinformation. |
| Bietresato et al. [71] | Volume reconstruction and mapping of vegetation. | (2D) Fusion of sonar, IMU and RTK GPS. | Plant volume calculation using LiDAR sensors and optical-data to obtain normalized difference vegetation index (NDVI) maps. | **Yes**. Mapping approach tested in indoor and outdoor environments. | Localization system not tested. | **Not tested.** | **Yes**. Does not require prior map information. |
| Utstumo et al. [72] | In-row weed control. | (2D) EKF-based. Uses a forward facing monocular camera and a GPS module. | Support Vector Machine (SVM) used to extract environment features and create a spray map. | **No**. Localization and mapping not tested. | Localization system not tested. | **Not tested.** | **Not tested.** |
| Santos et al. [8] (2020) | Autonomous navigation on steep slope vineyards. | (2D) PF-based. Fuses wheel odometry with a vision system. | Metric map composed of high-level landmarks detected using Deep Learning techniques. | **Yes**. One experiment done on a real vineyard. | Average error of 10.12 cm over the tested sequence. | **Not tested.** | **Yes**. The approach does not need apreviously built map. |

Autonomous navigation is being studied in tasks like precision agriculture in tree fruit production, greenhouse spraying; and in fields like steep slope vineyards, orchards, and olive groves. The implementation of autonomous robotic platforms in agriculture has been growing, in particular in tasks like weeding, seeding, disease and insect detection, crop scouting, spraying, and harvesting [73]. The localization algorithms of these works are, in their great majority, based on PFs, KFs (EKF and IF), or VO. From all the works collected, only Le et al. [65] developed a system able to localize the robot in 3D. This work uses a non-linear optimization technique based on the Lavenberg–Marquardt algorithm to solve for the robot pose. From the evaluation performed, we concluded that this work is the most suitable for autonomous navigation in agriculture, since it meets all the criteria: was tested in the real scenario, presents a high accuracy, scalability, and availability. Cheein et al. [66], also present a very interesting work. This is the only one based on an IF, and performs 2D localization with accuracy, availability and scalability. It is worth noting that the majority of works use either laser sensors, or visual sensors such as monocular or stereo camera systems. Wheel odometry is also commonly used to give filter-based approaches the control inputs. Few works use GNSS-based receivers, which shows the recurrent unavailability of satellite signals in agricultural environments, and so, the necessity of the creation of GNSS-free localization and mapping algorithms. In this context, the real time kinematic (RTK) satellite-based localization systems are rising up due to their high accuracy. In particular, Bietresato et al. [71] use this technology, fusing it with a sonar and an IMU to localize a robotic platform. This work is proposed in order to reconstruct the volume of plants and to map the vegetation, using laser data and NDVI information. On the negative side, the authors do not provide any experiments on the localization and mapping approach. VO methods tend to use only visual data, and have the advantage being, in theory, always available, in that they do not require prior mapping information. On the other hand, these families of localization methods are not, in general, scalable, since they accumulate error over the time, due to the integration of relative motions between image frames. As an example, Zaman et al. [61] propose a VO approach, tested in real scenarios such as soil, grass, and other terrains, that presents accurate results for short paths (0.08 mm error), but that degrades when increasing the path length. Similarly, Younse et al. [63] present a VO approach, only tested for a path with 305 cm extension, which does not prove to have practical availability for long-range motion types.

In terms of mapping, one can conclude that not all the methods perform this task. In particular, VO approaches do not map the environment. Furthermore, some works use localization-only approaches with prior information. For example, Bayar et al. [64] assume that the orchards rows length is known, and localization is performed relatively to the rows' lines. In other cases, mapping is performed offline [58–60]. In this context, Freitas et al. [58,59] perform an offline orchard metric mapping using artificial landmark detection, and use this information in the localization procedure. In this, points and line features are detected and matched to the prior map information to perform the robot pose estimation. From all the works collected, the one proposed by Chebrolu et al. [67] stands out as being the only one that performs semantic mapping. This work implements an offline mapping procedure, using aerial images to build a map of landmarks. Moreover, Santos et al. [8] use an interesting approach of computing 2D landmark location using deep learning techniques. In general, metric maps are used, by means of probabilistic occupancy grid maps, point clouds, or feature-based approaches.

From this study, many conclusions can be taken. Firstly, the fact that almost all the works were tested in real scenarios is exciting, in that it shows the ambition of researchers to actually implement autonomous navigation in agriculture. From these, we highlight [58,59] that perform tests in real sequences with more than 1.8 km each. In terms of accuracy, the majority of works present good results. For example, Le et al. [65] present a system with 2% translation error for a trajectory with approximately 500 m. Furthermore, Chebrolu et al. [67] propose a work that achieves a maximum error of 17 cm on a sequence with a path higher than 200 m. However, besides many of the works presenting acceptable performance, the problem of 3D localization in agriculture is still an area with

very low research impact. Only one work focuses on this problem, which shows that much work and development is yet to come. In particular, steep slope agricultural environments will require localization system able to accurately estimate all the 6-DoF of the robot pose. Additionally, for robots to operate safely in agricultural environments, longer term experiments should be carried out. This is due to the requirement of automatizing procedures during long periods, which imposes the need of long-term autonomous operability. In the same context, all-weather experiments are required, since the navigation systems should work under a vast range of meteorological conditions. In the current state-of-the-art, the majority of works focus on low- or mid-term experiments, and do not take into account the all-weather challenge. In addition, the mapping approaches are, in general, based on metric maps. Topological and semantic mapping are concepts almost nonexistent in this area. This also shows that much work can be done in this area. Semantic mapping can be important in the classification of agricultural agents, obstacle detection on these fields, etc. In this area, deep learning can have an important role, where high-qualified deep neural networks can be trained and used to provide an intelligent perception of the environment. All this information can be further used by human operators for example, for monitoring tasks, detection of anomalies, and others. Moreover, topological mapping can represent a huge advance in these area, when considering scalability. Global maps can be partitioned in local maps in a graph-like procedure, which can solve memory issues, allowing the robot to navigate safely during longer periods of time. Finally, not all the works present scalability and availability, which are concepts required before others which are more complex can arise such as recovery, updatability, or dynamicity.

## 5. Localization and Mapping in Forestry

Recent works approach the problem of localization and mapping in forestry. This work performed an intensive literature review in this area. Table 2 presents a summary of the works collected and their main characteristics.

**Table 2.** Summary of the works collected on localization and mapping related to the forestry field.

| Ref. | Forestry Application | Localization Approach | Mapping Application | Tested in Real Scenerio | Accuracy | Scalability | Availability |
|---|---|---|---|---|---|---|---|
| Qian et al. [74] (2016) | Accurate forest stem mapping. | (2D) Fusion of GNSS/INS with scan-match based approach solved using the Improved Maximum Likelihood Estimation. | Metric occupancy grid map built from laser scan data. | **Yes**. Real field experiments performed. | Positioning accuracy of 13 cm for the field data sequence. | **Yes**. Successful results in long-term sequence with 800 m. | **Yes**. The algorithm does not required prior map information. |
| Hussein et al. [75] (2015) | Autonomous navigation in forests. | (2D) Localization based on a scan-matching procedure. | Metric map of trees generated by on-board LiDAR sensors. Map matching with a global map generated from aerial orthoimagery. | **Yes**. Experiments performed on real forest. | Average error of <2 m for robot pose. | **Partially**. Long-term experiments performed, but with considerable errors. | **No**. Requires a map generation from aerial images. |
| Li et al. [76] (2020) | Autonomous harvesting and transportation. | (2D) Map matching localization approach based on Delaunay triangulation. | 3D LiDAR-based stem mapping. | **Yes**. Real experiment using a forestry dataset. | Location accuracy of 12 cm on the tested sequence. | **Yes**. Successful results in a long-term path (200 m). | **No**. Requires a previously built stem map. |
| Pierzchała et al. [77] (2018) | 3D forest mapping. | (3D) Graph-based SLAM. Uses the Levenberg-Marquardt method. | 3D point cloud map generated using LiDAR odometry, with graph optimization through loop closure detection. | **Yes**. Data recorded by authors' robot in a forest. | SLAM system provides tree positioning accuracy—mean error of 4.76 cm. | **Yes**. Successful long-term real experiments in sequence with 130.7 m. | **Yes**. The method performs online SLAM without need of prior map. |
| Rossmann et al. [78] (2013) | Autonomous navigation in forests. | (2D) PF-based. Fusion of a laser sensor and a GPS. | Offline generation of forest tree map. | **Yes**. However, no demonstration of results available. | Authors measure the location error in sample points in time. They claim to obtain mean error of 0.55 m. | **Not tested**. | **Not tested**. |
| Miettinen et al. [79,80] (2007) | Forest harvesting. | (2D) Feature-based SLAM. Computed using laser odometry. | Metric feature map, built by fusing laser data and GPS information. | **Yes**. Experiments on real outdoor environment. | **Not tested**, due to the unavailability of ground truth. | **Not tested**. | **No tested**. |
| Heikki et al. [81,82] (2013) | Stem diameter measure. | (3D) Laser-odometry approach, fused with an IMU. | Metric landmark map composed of stem detections and ground estimation. | **Yes**. Long-term real experiment performed (260 m). | 7.1 m error for a 260 m path. | **No**. High localization errors reported for a long-term path. | **Yes**. Does not need prior map information. |
| Tang et al. [83] (2015) | Biomass estimation of forest inventory. | (2D) Scan-matching based SLAM. Uses the Improved Maximum Likelihood Estimation algorithm. | Metric occupancy grid map built using laser data. | **Yes**. Long-term real experiment performed (300 m). | Obtained positioning error <32 cm in the real world experiment. | **Yes**. Successful performance in long-term experiment. | **Yes**. Does not need prior map information. |

The main goals for robotics in forest environments are stem mapping, harvesting, transportation, and biomass estimation. Similarly to the methods collected in the agricultural context, a great majority of the developed localization systems for forest environments only compute the robot pose in 2D. Only Pierzchała et al. [77], aiming to perform 3D mapping of a forest environment, propose a graph-based SLAM approach, that computes 6-DoF robot pose using the Levenberg–Marquardt algorithm. In addition, this work presents other contributions, such as built in-house datasets that were used to test and evaluate the SLAM pipeline. Moreover, Heikki et al. [81,82] propose a 3D SLAM algorithm where the translation components are calculated by a laser-odometry technique, and the rotational ones using an IMU. It is worth noting that the SLAM process also estimates the drifting bias of the yaw angle. On the negative side, this work reports high errors for long-term trajectories. All the other works propose 2D-based SLAM algorithms. From these, we highlight [74] that, in order to build accurate stem maps, fuses GNSS/INS with a scan-match approach solved using the improved maximum likelihood estimation method. This work presents an accuracy of 13 cm in a real experiment with 800 m, presenting scalability and availability, since it does not require prior mapping information. Furthermore, Li et al. [76] propose a very innovative localization approach based on map matching with Delaunay triangulation. Here, the Delaunay triangulation concept is used to match local observations with previously mapped tree trunks. This is an innovation since, unlike the majority of works that use point cloud-based matching approaches, the authors propose a topology-based method. This work achieved an accuracy of 12 cm in a long-term path with 200 m. On the negative side, it requires a previously built map so that it can operate. Similarly, Hussein et al. [75] propose a localization and mapping approach that requires a prior map. The interesting innovation of this work is the creation of a stem map using aerial orthoimagery. Thus, this approach does not require that the robot goes one first time to the field before it can perform autonomously. Miettinen et al. [79,80] use the interesting concept of laser odometry to localize the robot. This approach is related with VO in that it computes the relative transformation between consecutive sensor observations. The major difference is that, in laser odometry, a scan-matching approach is used to align consecutive observations and extract the relative transformation matrix. Moreover, using scan-matching concepts, Tang et al. [83] propose a 2D SLAM algorithm that solves the Improved maximum likelihood estimation method to localize the robot, with the goal of estimating the biomass forest inventory. Considering the overall collection of works, it is worth noting that visual sensors are not so used as in agriculture. Laser range sensor and GNSS-based receivers are the ones used in this field.

In terms of mapping, all the works perform this task, contrary to the case of agriculture. From all the works, we highlight [77], since it performs 3D point cloud map generated using LiDAR odometry, with graph optimization through loop closure detection. Moreover, only two works compute offline mapping. The first, proposed by Li et al. [76], requires a prior visit to the working environment, in order to create a global 3D stem map using a LiDAR sensor. In a more innovative way, Hussein et al. [75] build the global map using aerial orthoimagery, which enables the navigation with prior map information without mandatory previous visits to the working environment. In terms of mapping, another major conclusion is that metric maps play an important role in many SLAM algorithms. This is due to the frequent use of LiDAR sensors to build tree trunk (stem) maps. This approach is common since stems constitute highly trustworthy landmarks to use in SLAM algorithms. In this context, Miettinen et al. [79,80] use their SLAM algorithm not only to localize the robot, but also to build precise stem maps, with specific information of the forest structure. In this work, during the mapping procedure, tree diameters, positions and stand density are measured. In the same way, Heikki et al. [81,82] propose a mapping approach where the stem diameters are measured, and a ground model is built. This work reports a trunk diameter error of 4 cm for short range observations (less than 8 m). Given all this, we can conclude that, in forestry, mapping seems to be more evolved, with intense research focus on stem detection and mapping.

From this collection of works, many conclusion can be taken. After an intensive search, only nine proper works were found, and most of them were proposed after 2013. This shows that this

research line is still quite new, not yet developed, and has many open topics. From these topics, we can highlight the issue of availability. From the reported works, only half of them showed the ability of working without prior map information, which shows that online SLAM is still under-developed in this field. Moreover, as mentioned before, 3D localization is still quite rare in forest environments. This can deny autonomous navigation in steep forests with considerable inclinations, and different levels of altimetry. On the other side, mapping seems a more advanced area, with some works focused on creating 3D maps of the forests, and others developing methods for stem mapping. Even so, just like in the agricultural field, metric maps are quite abundant. So, more advanced mapping concepts such as semantic and topological perception of the environment are still quite under-developed. Moreover, many works propose the detection of tree trunks to use as landmarks in the localization procedure. However, none of them currently use deep learning concepts to detect such agents in the forest fields. Since the tree trunk diameter measurement is one of the fields of interest in forestry, semantic segmentation can play an important role in the future. These algorithms use neural networks to segment images or point clouds into semantic objects, at the pixel or point level. The use of visual sensors not common in forestry is also related to this. The implementation of visual perception algorithms can also improve the localization and mapping procedures. On the bright side, just like in agriculture, the works were tested in real scenarios, which shows that resources are being channeled to this research field.

## 6. Datasets for Localization and Mapping in Agriculture and Forestry

In order to have reliable data to develop new concepts and algorithms related with localization and mapping in agriculture and forestry, the creation of open-source datasets is essential. Table 3 shows a collection of 5 works done in this area, all of them developed after 2016.

Chebrolu et al. [84] propose an agricultural dataset for plant classification and robotics navigation on sugar beet fields. This work provides a variety of sensors, such as RGB-D cameras, 3D LiDAR sensors, GPS, and wheel odometry, all calibrated extrinsically and intrinsically between each other. The main interesting feature of this work was the recording of data during a period of three months, which can be used to test localization and mapping approach with long-term considerations. Kragh et al. [85] propose a collection of raw data from sensors mounted on a tractor operating in a grass mowing scenario. This work provides precise ground-truth localization using a fusion of IMU and GNSS-based receiver. The main limitation of this proposal is the lack of data from different times of the day, or year. In [86] a dataset for test and evaluation of visual SLAM approaches in forests is presented. In this, data are provided from four RGB cameras, an IMU, and a GNSS receiver, all of them calibrated and synchronized. Data were recorded in summer and winter seasons, and different times of the day. In [87] is presented a dataset with six sequences in soybean fields. This works has in consideration repetitive scenes, reflection, rough terrains, and other harsh conditions. Data were collected on two different days, but at the same period of the day. Finally, Reis et al. [88] present a set of datasets with information from a variety of sensors, suitable for the localization and mapping purposes. In this work, a state-of-the-art SLAM approach was tested under the different datasets recorded in different forests. Overall, all the methods present a large-scale dataset, with long distances travelled. In the near future, with the intense growth of deep learning models in robotic systems, more advanced datasets should be created. In particular, sensor data can be put together with object labelling. In this context, visual data can be provided along with landmark annotation, both at object and pixel level. Furthermore, 3D LiDAR data can also be annotated in order to enable researchers to train and test their own semantic segmentation models.

**Table 3.** Summary of the works collected presenting datasets for use in autonomous navigation in agriculture and forestry.

| Ref. | Description | Large-Scale | Long-Term |
|---|---|---|---|
| Chebrolu et al. [84] (2017) | Agricultural dataset for plant classification and robotics navigation on sugar beet fields. Provides data from RGB-D camera, 3D LiDAR sensors, GPS, and wheel odometry. All the sensors are calibrated extrinsically, and intrinsically. | **Yes**. | **Yes**. Recorded over a period of three months, and, on average, three times a week. |
| Kragh et al. [85] (2017) | Raw sensor data from sensors mounted on a tractor in a grass mowing scenario. It includes stereo camera, thermal camera, web camera, 360° camera, LiDAR, and radar. Precise vehicle localization obtained from fusion of IMU and GNSS. | **Yes**. Data recorded on a large field with 2 ha. | **No**. The dataset has approximately 2 h, all at the same day. |
| Ali et al. [86] (2020) | Dataset for visual SLAM on forests. The vehicle is equipped with four RGB cameras, an IMU, and a GNSS receiver. Sensor data is calibrated and synchronized. | **Yes**. Range of distance travelled varies from 1.3 km to 6.48 km. | **Yes**. Data recorded on summer and winter conditions. Also, different times of the day were considered. |
| Pire et al. [87] (2019) | Dataset with six sequences in soybean fields. Considers harsh conditions such as repetitive scenes, reflection, rough terrain, etc. Contains data from wheel odometry, IMU, stereo camera, GPS-RTK. | **Yes**. Total length trajectory around 2.3 km. | **Partially**. Data recorded in two separate days, but in the same time of the year, and the day. |
| Reis et al. [88] (2019) | Dataset containing data from different sensors such as 3D laser data, thermal camera, inertial units, GNSS, and RGB camera in forest environments. | **Yes**. Data recorded in three different large-scale forests. | No information. |

## 7. Conclusions

This work proposed the analysis of the current state-of-the art of localization and mapping in agriculture and forestry. After an intensive search, 15 works were collected related to localization and mapping in agriculture, 9 were collected in the context of forestry, and 5 works that present datasets for test and evaluation of these approaches were characterized. The works were characterized in terms of their agricultural/forestry application, the localization and mapping approaches, and their accuracy, availability and scalability. This study leads to the main conclusion that this research line is still premature, and many research topics are still open. In particular, 3D localization is quite rare in these environments. Furthermore, advanced mapping techniques that are now present in other areas such as topological and semantic mapping are yet to develop in agriculture and forestry. This being said, we believe that this area has a lot of potential to grow, and that, in the near future, many works and innovations will arise to bridge the current faults of the state-of-the-art.

**Author Contributions:** Conceptualization, A.S.A. and F.N.d.S.; methodology, A.S.A. and F.N.d.S. and J.B.C.; validation, F.N.d.S. and J.B.C. and H.S. and A.J.S., F.N.d.S. and J.B.C., H.S., A.J.S.; investigation, A.S.A. and F.N.d.S.; resources, F.N.d.S. and J.B.C.; writing—original draft preparation, A.S.A.; writing—review and editing, F.N.d.S. and J.B.C.; supervision, F.N.d.S. and J.B.C. and H.S. and A.J.S. All authors have read and agreed to the published version of the manuscript.

**Funding:** This work is funded by funds through the FCT—Fundação para a Ciência e a Tecnologia, I.P., within the framework of the project "WaterJPI/0012/2016". The authors would like to thank the EU and FCT for funding in the frame of the collaborative international consortium Water4Ever financed under the ERA-NET Water Works 2015 cofounded call. This ERA-NET is an integral part of the 2016 Joint Activities developed by the Water Challenge for a changing world joint programme initiation (Water JPI).

**Conflicts of Interest:** The authors declare no conflict of interest.

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
