# Peer review of "Localization and Mapping for Robots in Agriculture and Forestry: A Survey"

_robotics, doi:10.3390/robotics9040097_

Round 1

Reviewer 1 Report

This paper proposed the analysis of the current Localisation and Mapping methods in agriculture and forestry. The work first reviewed a number of SLAM algorithms and then compared related works. The review and the comparison of this paper were too general. As a survey paper, the reviews should be deeper.

Author Response

Dear reviewer,

We are truly grateful for your comments and reviews. All of them helped to improve the article quality.

Attached we send our responses to your comments.

Sincerely,

André Aguiar.

Reviewer 2 Report

The paper presents a survey on localization and mapping for mobile robots in the agriculture and forestry fields. The paper is well written, interesting and easy to read. However, the following points need to be addressed to improve the quality of the work:

1) At the beginning of section 2.1.2. Mapping the environment, I suggest the authors to better explain which are the on-board sensors that are used for the mapping of the environment by the robotic platform.

2) I suggest to place large tables with landscape orientation. In this manner, a larger font size can be used to enhance the readability of the text.

3) The following references should be considered to improve the literature review:

Habibie, N., Nugraha, A. M., Anshori, A. Z., Ma'sum, M. A., & Jatmiko, W. (2017, December). Fruit mapping mobile robot on simulated agricultural area in Gazebo simulator using simultaneous localization and mapping (SLAM). In 2017 International Symposium on Micro-NanoMechatronics and Human Science (MHS) (pp. 1-7). IEEE.

Bietresato, M., Carabin, G., D'Auria, D., Gallo, R., Ristorto, G., Mazzetto, F., Vidoni, R., Gasparetto, A., Scalera, L., (2016, August). A tracked mobile robotic lab for monitoring the plants volume and health. In 2016 12th IEEE/ASME International Conference on Mechatronic and Embedded Systems and Applications (MESA) (pp. 1-6). IEEE.

Iqbal, J., Xu, R., Sun, S., & Li, C. (2020). Simulation of an Autonomous Mobile Robot for LiDAR-Based In-Field Phenotyping and Navigation. Robotics9(2), 46.

Fountas, S., Mylonas, N., Malounas, I., Rodias, E., Hellmann Santos, C., & Pekkeriet, E. (2020). Agricultural Robotics for Field Operations. Sensors20(9), 2672.

Author Response

Dear reviewer,

We are truly grateful for your comments and reviews, which helped to improve the article's quality.

We attach a file with the responses to your suggestions.

Sincerely,

André Aguiar.

Round 2

Reviewer 1 Report

All reviewers' comments are addressed accordingly, and the quality of the paper has been improved. The reviewer suggests accepting the paper to be published. 

Reviewer 2 Report

The quality of the paper has been improved with respect to the previous version. I suggest the paper to be accepted for publication.